# Genome-Wide Identification, Evolution, and Expression Analyses of AP2/ERF Family Transcription Factors in *Erianthus fulvus*

**DOI:** 10.3390/ijms24087102

**Published:** 2023-04-12

**Authors:** Zhenfeng Qian, Xibing Rao, Rongqiong Zhang, Shujie Gu, Qingqing Shen, Huaying Wu, Shaozhi Lv, Linyan Xie, Xianli Li, Xianhong Wang, Shuying Chen, Lufeng Liu, Lilian He, Fusheng Li

**Affiliations:** 1College of Agronomy and Biotechnology, Yunnan Agricultural University, Kunming 650201, China; 2Sugarcane Research Institute, Yunnan Agricultural University, Kunming 650201, China; 3The Key Laboratory for Crop Production and Smart Agriculture of Yunnan Province, Kunming 650201, China

**Keywords:** *Erianthus fulvus*, *AP2/ERF* gene family, genome-wide, abiotic stress, expression pattern

## Abstract

The AP2/ERF transcription factor family is one of the most important gene families in plants and plays a vital role in plant abiotic stress responses. Although *Erianthus fulvus* is very important in the genetic improvement of sugarcane, there are few studies concerning *AP2/ERF* genes in *E. fulvus*. Here, we identified 145 *AP2/ERF* genes in the *E. fulvus* genome. Phylogenetic analysis classified them into five subfamilies. Evolutionary analysis showed that tandem and segmental duplication contributed to the expansion of the EfAP2/ERF family. Protein interaction analysis showed that twenty-eight EfAP2/ERF proteins and five other proteins had potential interaction relationships. Multiple cis-acting elements present in the *EfAP2/ERF* promoter were related to abiotic stress response, suggesting that *EfAP2/ERF* may contribute to adaptation to environmental changes. Transcriptomic and RT-qPCR analyses revealed that *EfDREB10*, *EfDREB11*, *EfDREB39*, *EfDREB42*, *EfDREB44*, *EfERF43*, and *EfAP2-13* responded to cold stress, *EfDREB5* and *EfDREB42* responded to drought stress, and *EfDREB5*, *EfDREB11*, *EfDREB39*, *EfERF43*, and *EfAP2-13* responded to ABA treatment. These results will be helpful for better understanding the molecular features and biological role of the *E. fulvus AP2/ERF* genes and lay a foundation for further research on the function of *EfAP2/ERF* genes and the regulatory mechanism of the abiotic stress response.

## 1. Introduction

Adverse environmental conditions, such as low temperature and drought, severely affect plant growth and survival. However, after a long period of acclimation, plants have gradually acquired response mechanisms for coping with these stresses. Plants adapt to adverse environmental conditions by sensing external stress signals, resulting in specific responses via changes in gene expression, metabolism, and physiological traits [1,2]. Among these complex processes, the expression of transcription factors (TFs) and the transcription of downstream responsive genes are key to plant resistance to stress conditions [3,4,5]. The AP2/ERF (APETALA2/ethylene response factor) family is one of the largest TF families in plants. Members of the AP2/ERF family contain the highly conserved AP2 DNA-binding domain, which consists of 60–70 amino acids [6]. To date, AP2/ERF family TFs have been identified and analyzed in a number of species, such as *Arabidopsis thaliana* [7], *Oryza sativa* [6], *Saccharum spontaneum* [8], *Hordeum vulgare* [9], *Triticum durum* [10], and *Zea mays* [11]. However, the function of only a few AP2/ERF TFs has been revealed.

Many AP2/ERF TFs have been reported to participate in plant growth and development regulation [12], biotic and abiotic stress responses [13,14,15], and hormone responses [16,17]. For instance, constitutive expression of *OsEREBP1* in rice activates the jasmonic acid (JA) and abscisic acid (ABA) signaling pathways and enhances tolerance to *Xanthomonas oryzae pv. oryzae* (Xoo) and drought stress [18]. Apple *MdDREB2* positively regulates ABA biosynthesis by activating *mdNCED6/9* gene expression [17]. *NTRAV-4* enhances tolerance to drought stress by improving antioxidant defense capacity and cell membrane stability in tobacco [19]. Rice *OsERF3* promotes adventitious root development [20]. *Malus baccata* L. *MbERF12* depends on ethylene signal mediation to improve reactive oxygen species (ROS) scavenging ability in *Arabidopsis*, and plays a key role in cold-stress response [21]. *HcTOE3*, from the AP2 subfamily of *Halostachys caspica*, enhances tolerance to cold stress by upregulating the transcription levels of cold response genes (*CBF1*, *CBF2*, *COR15*, *COR47*, *KIN1*, and *RD29A*) and ABA signal transduction pathway genes (*ABI1*, *ABI2*, *ABI5*, and *RAB18*) in *Arabidopsis* [22]. These studies demonstrate that AP2/ERF family TFs play an important role in plant responses to low-temperature and drought stresses. Therefore, it is important to explore AP2/ERF TFs in plants and reveal their stress resistance function and regulatory mechanism.

Sugarcane (*Saccharum* spp. Hybrid) contributes to 80% of world sugar production and 40% of biofuel production, and is an important commercial crop [23]. Low temperature and drought are the main environmental factors affecting sugarcane growth, yield, and quality [24,25]. Therefore, mining sugarcane germplasm resources with strong cold and drought resistance to provide parent or donor genes for sugarcane breeding has become the main focus of stress-resistant sugarcane breeding. *Erianthus fulvus* Ness. (Chromosome, 2n = 20), a species closely related to sugarcane, has good cold tolerance and drought resistance characteristics and is a valuable germplasm resource material for sugarcane breeding and research [26]. However, the mechanisms of cold tolerance and drought resistance in *E. fulvus* are still unclear.

In a previous study, we demonstrated that the *AP2/ERF* family gene *EfDREB1A* plays an important role in the *E. fulvus* response to cold stress [27]. However, the classification, evolution, expression pattern, regulatory mechanism, and biological function of AP2/ERF family TFs in *E. fulvus* are still unclear. In this study, we systematically identified AP2/ERF family TFs in *E. fulvus*. The phylogenetic relationships, gene structure, conserved domains, promoters, chromosomal location distribution, gene duplication, and protein interactions were subsequently analyzed. In addition, the effects of *AP2/ERF* genes on *E. fulvus* adaptation to cold and drought stresses and ABA treatment were analyzed to enhance our understanding of their biological functions. This study provides valuable information for further revealing the functions and regulatory mechanisms of *AP2/ERF* family genes in *E. fulvus*.

## 2. Results

### 2.1. Identification and Classification of EfAP2/ERF Genes

A total of 145 complete *AP2/ERF* family genes were identified in *E. fulvus* (Appendix A). The predicted genes ranged from 401 to 19,478 bp in length and encoded proteins with 133–2463 amino acids (aa). The molecular weight (MW) and theoretical isoelectric point (pI) range of EfAP2/ERF family proteins were 14,745.9–274,687.6 Da and 4.26–12.64, respectively. Based on the number of conserved AP2 domains and amino acid sequence similarities, the *EfAP2/ERF* genes were divided into five subfamilies, namely, *DREB*, *ERF*, *AP2*, *RAV*, and *Soloist*. The DREB subfamily contains 48 genes (*EfDREB1* to *EfDREB48*) that encode proteins with one conserved AP2 domain. The ERF subfamily comprises 75 members (*EfERF1* to *EfERF75*) with one conserved AP2 domain. The AP2 subfamily has sixteen genes (*EfAP2-1* to *EfAP2-16*), of which fourteen have two conserved AP2 domains, and two have only one conserved AP2 domain. The RAV subfamily contains three genes (*EfRAV1* to *EfRAV3*), which encode proteins with conserved AP2 and B3 domains. The Soloist subfamily consists of three genes (*EfSoloist1* to *EfSoloist3*) that encode proteins with an AP2-like domain. The total number of *AP2/ERF* family genes in *E. fulvus* was close to that in *Arabidopsis* (146) [7], rice (163) [6], and *sorghum* (126 ERF genes) [28]. However, due to the large size of the genome, there are 218 AP2/ERF family genes in *S. spontaneum* [8].

### 2.2. Phylogenetic Analysis of AP2/ERF Genes

To characterize the evolutionary relationships of *EfAP2/ERF* genes, phylogenetic trees of EfAP2/ERF protein sequences were constructed based on multiple sequence alignment (Figure 1). Similar to the clustering of *Arabidopsis* [7], sorghum [28], and *S. spontaneum* [8], the phylogenetic trees clustered all of the EfAP2/ERF proteins into five subfamilies (DREB, ERF, AP2, RAV, and Soloist). In all of the EfDREB subfamily proteins, position 14 in the AP2 domain is valine (V), whereas position 14 in the AP2 domain of most of the EfERF subfamily proteins is alanine (A) (Appendix A), which is similar to the results in rice [6]. According to the classification criteria in *Arabidopsis* [7] and rice [6], the EfDREB and EfERF subfamilies were further divided into five groups (A1 to A2 and A4 to A6) and six groups (B1 to B6), respectively (Figure 2). Groups A1 to A2 and A4 to A6 encode EfDREB proteins, with 15, 3, 12, 9, and 9 members, respectively. The groups B1 to B6 encode EfERF proteins, with 15, 11, 20, 14, 8, and 7 members, respectively. However, members of the A3 group were absent in *E. fulvus*.

### 2.3. Conserved Motif and Gene Structure Analysis

To characterize the diversity of *EfAP2/ERF* family genes, we performed prediction and analysis of conserved motifs (Figure 3, Appendix A). A total of 25 conserved motifs (motif1 to motif25) were predicted in the EfAP2/ERF family. Motif 1 and motif 2 were found in all members of the DREB and ERF subfamilies. The characteristics of motif4-motif2-motif1-motif3 were detected in most of the DREB and ERF subfamilies (Figure 3a,b). Motif 8, motif 9, and motif 15 appeared only in groups A6, B2, and B5, respectively. In addition, the characteristics of motif14-motif9/2-motif6-motif3-motif5-motif4-motif10 were unique to the AP2 subfamily, and motifs 18 and 12 were detected in only the RAV subfamily (Figure 3c). Similar to the clustering results from the phylogenetic trees, the motifs were almost similar on the same branch, suggesting that EfAP2/ERF proteins within the same clade may have similar functions.

To further explore the structural features of *EfAP2/ERF* family genes, the gene structure was investigated (Figure 4). Twenty-eight percent of the *EfAP2/ERF* family genes contained untranslated region (UTR) sequences, including 9 *EfDREB* genes, 21 *EfERF* genes, 6 *EfAP2* genes, 3 *EfRAV* genes, and 1 *Soloist* gene. A total of 78 *AP2/ERF* family genes had no introns. Similar to the structure of *DREB* subfamily genes in *S. spontaneum* [8], most of the *EfDREB* genes (34 genes) in *E. fulvus* had no introns (Figure 4a). In addition, 35 *EfERF* genes had introns (Figure 4b); all genes of the EfAP2 subfamily had multiple introns. All genes of the EfRAV subfamily had no introns, and two members of the EfSoloist subfamily had introns (Figure 4c). These results indicated that the gene structures were diverse among different subfamilies.

### 2.4. Chromosome Distribution

To characterize the location of *EfAP2/ERF* genes on the chromosome, we analyzed the chromosomal distribution of *AP2/ERF* genes. The 144 *EfAP2/ERF* genes were unevenly distributed on 10 chromosomes of *E. fulvus* (Figure 5). Most of the *EfAP2/ERF* genes were distributed on chromosomes Chr1, Chr3, Chr4, Chr5, Chr6, Chr7, Chr8, and Chr9, which harbored 16, 21, 16, 24, 21, 14, 12, and 10 genes, respectively, while Chr2 and Chr10 had only seven and three genes, respectively.

### 2.5. Duplication and Synteny Analysis of AP2/ERF Genes

Numerous studies have indicated that gene duplication events, including tandem duplication, segmental duplication, and whole-genome duplication (WGD), are a major driving force in the expansion of gene families [29,30]. To explore the duplication events of *EfAP2/ERF* genes, we performed tandem and segmental duplication analysis in *E. fulvus* (Figure 5 and Figure 6, Appendix A). A total of nine pairs of sixteen tandem duplication genes were identified, which were located on chromosomes Chr1, Chr5, Chr6, Chr7, and Chr8 (Figure 5). These tandem duplication genes included members of the EfDREB subfamily, with three pairs (*EfDREB18*-*EfDREB19*, *EfDREB22*-*EfDREB23*, *EfDREB32*-*EfDREB33*), and members of the EfERF subfamily, with six pairs (*EfERF4*-*EfERF5*, *EfERF47*-*EfERF48*, *EfERF48*-*EfERF49*, *EfERF49*-*EfERF50*, *EfERF61*-*EfERF62*, *EfERF67*-*EfERF68*). In addition, we found that the expansion of the EfAP2/ERF family was mainly derived from segmental duplication. A total of 30 pairs of 58 segmental duplication genes were identified on chromosomes (Figure 6). These segmental duplication genes included members of the EfDREB subfamily (with 16 pairs), EfERF subfamily (with 11 pairs), EfAP2 subfamily (with two pairs), and EfRAV subfamily (with one pair).

The divergence times of *EfAP2/ERF* genes with synteny were further estimated based on Ks values (Appendix A). The divergence time of the *EfAP2/ERF* tandem duplication gene pairs ranged from 8.08 to 54.06 million years ago (Mya), indicating that these gene pairs were formed via recent gene duplication events in *E. fulvus*. The divergence times of segmental duplication gene pairs were 24.30–296.15 Mya. Thirteen gene pairs arose 24.30–54.75 Mya, indicating that these gene pairs arose from recent gene duplication events. Fourteen gene pairs ranged arose 63.87 to 159.07 Mya, indicating that these gene pairs were formed via early gene duplication events. Three gene pairs were ancient, with divergence times ranging from 240.38 to 296.15 Mya.

In addition, the nonsynonymous-to-synonymous substitution rate ratio (Ka/Ks) was calculated to examine the selection type of *EfAP2/ERF* genes with synteny during evolution (Appendix A). The Ka/Ks ratio of *EfAP2/ERF* tandem duplication gene pairs ranged from 0.22 to 2.03, and these gene pairs, with a Ka/Ks ratio < 1, accounted for 55.56% of the gene pairs tested. The Ka/Ks ratio for *EfAP2/ERF* segmental duplication gene pairs ranged from 0.059 to 2.05, and 70% of the gene pairs had Ka/Ks < 1. A total of 66.67% of the *EfAP2/ERF* gene pairs in the *E. fulvus* genome had a Ka/Ks < 1, indicating that the *EfAP2/ERF* genes may have been subjected to purifying selection pressure during evolution. Furthermore, 33.33% of the *EfAP2/ERF* gene pairs had a Ka/Ks > 1, which indicated that these genes may have been subjected to positive selection after duplication.

### 2.6. Evolutionary Analysis of AP2/ERF Genes between E. fulvus and Other Species

To evaluate the evolutionary origin of the *EfAP2/ERF* genes, we performed a synteny analysis between *E. fulvus* and the other four species (*S. spontaneum*, sorghum, rice, and *Arabidopsis*) (Figure 7, Appendix A). There were 11 syntenic gene pairs between *E. fulvus* and *Arabidopsis* and 107, 131, and 136 between *E. fulvus* and rice, sorghum, and *S. spontaneum*, respectively. Moreover, a total of 45 *EfAP2/ERF* genes were synonymous with genes in rice, sorghum, and *S. spontaneum*, indicating that the AP2 family was highly conserved in *Gramineae*. Interestingly, the syntenic gene pairs of *E. fulvus* and sorghum were densely distributed on chromosomes, indicating that there was high homology between genes from *E. fulvus* and sorghum. In addition, the syntenic gene pairs of *E. fulvus* and *S. spontaneum* were also highly homologous, and each chromosome of *E. fulvus* was homologous to the four chromosomes of *S. spontaneum*, which suggested a sister relationship between *E. fulvus* and *S. spontaneum*, and that the chromosome of *S. spontaneum* may have grown through rearrangement.

### 2.7. Analysis of Putative Cis-Acting Regulatory Elements in EfAP2/ERF Promoters

Cis-acting elements are essential for understanding the expression differences and biological functions of these *EfAP2/ERF* genes. We analyzed the promoter sequences of *EfAR2/ERF* genes to identify possible cis-acting elements (Figure 8, Appendix A). Our analysis found that 18 types of cis-acting elements involved in abiotic stress-related, hormone-response, transcription factor binding, and developmental process-related were present in the promoter region of these *EfAP2/ERF* genes. The five subfamilies of genes had similar cis-acting element types, but the number of copies of each cis-acting element differed. Among these cis-acting elements, ABA-responsive, MeJA-responsive, low-temperature-responsive, and light-responsive elements were the most common, suggesting that *EfAP2/ERF* genes might have potential functions involved in hormone signal transduction and abiotic stress response. In addition, binding sites of various TFs, including DRE cis-acting elements, Myb-binding sites, WRKY-binding site elements, and MYC-binding site elements, were also common, indicating that these TFs may participate in the regulation of *EfAP2/ERF* gene expression.

### 2.8. Interaction Network of EfAP2/ERF Proteins

To investigate the interaction relationship between EfAP2/ERF and other proteins in *E. fulvus*, we constructed an interaction network based on the interolog of the network in *Arabidopsis* (Figure 9, Appendix A). A total of 68 proteins were mapped to the interaction network, of which 33 were in the EfAP2/ERF family and 5 belonged to other protein families. The results suggest that the functions of some EfAP2/ERF proteins may depend on interactions with other proteins.

### 2.9. Expression Pattern of EfAP2/ERF Genes in Response to Cold Stress

To understand the expression patterns of *EfAP2/ERF* genes in different tissues and under cold stress, we analyzed the expression profiles of *EfAP2/ERF* family genes using RNA-seq data of *E. fulvus* (Figure 10 and Appendix A). The *EfAP2/ERF* genes were differentially expressed in different tissues and under low-temperature stress. A total of 35 *EfDREB* genes were upregulated by cold stress induction, among which 16 genes, 7 genes, and 12 genes were highly expressed in leaves, stems, and roots, respectively (Figure 10a). Most of the *EfERF* genes were cold stress-inducible genes, among which 28 genes, 14 genes, and 17 genes were highly expressed in roots, stems, and leaves, respectively (Figure 10b). A total of 10 *EfAP2* genes were upregulated by cold stress induction, among which four genes and six genes were highly expressed in roots and stems, respectively. In addition, *EfRAV1* and *EfRAV3* were found to be upregulated in leaves and stems under cold stress (Figure 10c). These results indicated that the expression of *EfAP2/ERF* genes is tissue-specific, and *EfDREB*, *EfERF*, and *EfAP2* genes may participate in the cold stress response primarily in leaves, roots, and stems, respectively. Interestingly, nine *EfAP2/ERF* genes, including seven *DREB* genes (*EfDREB5*, *EfDREB10*, *EfDREB11*, *EfDREB16*, *EfDREB39*, *EfDREB42* and *EfDREB44*), one *ERF* gene (*EfERF43*) and one *AP2* gene (*EfAP2-13*), were significantly upregulated in leaves under cold stress (Appendix A), indicating that these genes play an important role in the cold stress response.

To confirm whether *EfAP2/ERF* gene expression was affected by low-temperature stress, we examined the expression levels of nine selected genes (*EfDREB5*, *EfDREB10*, *EfDREB11*, *EfDREB16*, *EfDREB39*, *EfDREB42*, *EfDREB44*, *EfERF43*, and *EfAP2-13*) in *E. fulvus* leaves by RT-qPCR (Figure 11). Consistent with the RNA-seq results, the seven *EfAP2/ERF* genes (*EfDREB10*, *EfDREB11*, *EfDREB39*, *EfDREB42*, *EfDREB44*, *EfERF43*, and *EfAP2-13*) were significantly upregulated under cold stress, suggesting that they may have critical roles in cold stress adaptation. However, the expression levels of two genes, *EfDREB5* and *EfDREB16*, were inconsistent with the results of RNA-seq and were not significantly upregulated after cold treatment.

### 2.10. Expression of EfAP2/ERF Genes in Response to Drought Stress and ABA Treatments

Many studies have indicated that *DREB* genes play an important role in plant responses to drought stress [31,32]. We further examined the expression levels of these nine genes under drought stress (Figure 12). *EfDREB42* expression was also enhanced after 9 d of drought stress (fold-change > 2.2), suggesting that *EfDREB42* might play significant roles in the response to drought stress. In contrast to the expression pattern under low-temperature stress, the expression levels of six genes (*EfDREB10*, *EfDREB11*, *EfDREB39*, *EfDREB44*, *EfERF43*, and *EfAP2-13*) showed no significant up- or downregulation under drought stress, suggesting that these genes may not be involved in the drought stress response. In addition, *EfDREB16* also exhibited no significant changes. In contrast, *EfDREB5* was upregulated under drought stress, and the fold upregulation (>3.5) was significant at 9 d.

Many studies have indicated that *DREB/CBFs* participate in the response to abiotic stress via both ABA-dependent and ABA-independent pathways [33,34]. As ABA response elements were the most common in the promoter regions of *EfAP2/ERF* genes, we further analyzed the expression levels of nine *EfAP2/ERF* genes following ABA treatments (Figure 13). After ABA treatment, four *EfAP2/ERF* genes, namely, *EfDREB11*, *EfDREB39*, *EfERF43,* and *EfAP2-13*, were shown to be significantly downregulated. In contrast, *EfDREB5* was upregulated, and the fold upregulation (>2.4) was significant at 12 h. In addition, four *EfDREB* genes (*EfDREB10*, *EfDREB16*, *EfDREB42*, and *EfDREB44*) showed no significant up- or downregulation after ABA treatment. These results suggest that five *EfAP2/ERF* genes (*EfDREB5*, *EfDREB11*, *EfDREB39*, *EfERF43*, and *EfAP2-13*) and three *EfDREB* genes (*EfDREB10*, *EfDREB42*, and *EfDREB44*) might participate in the response to abiotic stress via ABA-dependent and ABA-independent pathways, respectively.

## 3. Discussion

As one of the largest and most important TF families in plants, the *AP2/ERF* family plays a vital role in regulating plant growth and development and in biotic and abiotic stress tolerance [35,36,37]. To date, the *AP2/ERF* gene family has been extensively studied in some plants (such as *Arabidopsis*, rice, maize, and cotton) [7,38,39,40]. However, due to singularly complex genomes, there is still little information concerning the structure, evolution, and function of *AP2/ERF* genes in the *Saccharum* complex. Although the AP2/ERF family has been identified in *S. spontaneum* [8], the functions of these genes are still unclear. In a previous study, we uncovered the cold stress-induced gene *EfDREB1A* in *E. fulvus* [27]. To further investigate the AP2/ERF family in *E. fulvus*, a total of 145 *EfAP2/ERF* genes (Appendix A) were identified from the *E. fulvus* genome database [41]. The number of *EfAP2/ERF* genes in *E. fulvus* was close to that in *Arabidopsis* (146, genome size: 125 Mb) [7], rice (163, genome size: 466 Mb) [6], and sorghum (126, genome size: 750 Mb) [28]. Due to the small size of the genome of *E. fulvus* (chromosome: 2n = 20; genome size: 0.9 Gb) [41], the number of *EfAP2/ERF* genes was fewer than that in *S. spontaneum* (218, genome size: 3.36 Gb) [8], and maize (292, genome size: 2.3 Gb) [11]. These results suggest that the number of *AP2/ERF* family genes in different plants varies due to species specificity and variations in genome size.

AP2/ERF TFs are highly conserved and widespread in plants. Each member of the AP2/ERF family contains the typical conserved AP2 domain, which is composed of approximately 50–70 amino acids [42,43]. Based on sequence similarities and the number of AP2 domains, phylogenetic analysis divided the *EfAP2/ERF* genes into five subfamilies (Figure 1): EfERF (with one conserved AP2 domain), EfDREB (with one conserved AP2 domain), EfAP2 (with one or two conserved AP2 domains), EfRAV (with conserved AP2 and B3 domains), and EfSoloist (with one AP2-like domain) [6,8]. Although the EfDREB and EfERF subfamilies both have one AP2 domain, they exhibit certain differences: the amino acid at position 14 in the AP2 domain of the EfDREB and EfERF subfamilies is V and A, respectively (Appendix A). This basis of classification was very important and consistent with the results in rice [6].

The conserved motifs are important elements of functional domains [44]. The AP2 domain is highly conserved in all plants [45]. Motif 1 located on the AP2 domain contains WLG and RAYD elements in the EfAP2/ERF family, which is very important for the structure or function of these proteins [9,11,46]. Previous studies revealed that the LWS (I/L/Y) element exists in the DREB (A1 and A5 groups) and ERF (B2 group) subfamilies in *Arabidopsis*, rice, and sorghum [28,44]. In this study, we also found that the EfDREB (A1 and A5 groups) and EfERF (B2 group) subfamilies contained LWS (F/Y) elements (located in motif 25) (Figure 3 and Appendix A). These results further indicated that the AP2/ERF family is conserved in plants. Moreover, similar to the clustering results from the phylogenetic trees, the motifs were similar on the same branch, suggesting that EfAP2/ERF proteins within the same clade may have similar functions.

Gene families evolve from a primitive ancestor. Gene duplication events generate new homologous genes and lead to gene family functional divergence, which plays a key role in species evolution, genome amplification, and gene family expansion [8]. Many studies have demonstrated that tandem, whole-genome, and segmental duplications are the main pathways for gene family expansion [8,10,47,48]. In this study, we found that the expansion of the *EfAP2*/*ERF* gene family was derived from tandem duplication and segmental duplication events. A total of nine tandem duplication collinear gene pairs were generated by recent gene duplication events; thirty segmental duplication collinear gene pairs arose via ancient, early, and recent gene duplication events (Figure 5 and Figure 6, Appendix A). In addition, *EfAP2/ERF* family genes were under intense purification selection pressure during evolution and may have maintained their functional stability [8,10,47,48]. Previous studies revealed that eukaryotes have experienced the process from gain to loss of introns during evolution [49]. The *DREB* and *RAV* subfamilies of *E. fulvus* (Figure 4) and *S. spontaneum* seem to have experienced intron loss [8].

Synteny analysis can identify chromosome structural changes and homologous gene evolutionary and functional connections in multiple genomes [50,51]. In this study, synteny analysis showed that the collinear gene pairs of *E. fulvus* and sorghum were densely distributed on chromosomes and had one-to-one correspondence, indicating that the genes *E. fulvus* and sorghum were homologous, and both of these species may have emerged in the same period of evolution (Figure 7). Moreover, the *AP2/ERF* family genes may also have been subjected to similar environmental selection during the evolution process. In addition, the syntenic gene pairs of *E. fulvus* and *S. spontaneum* were also highly homologous and numerous, and each chromosome of *E. fulvus* was homologous to the four chromosomes of *S. spontaneum*, which suggested a sister relationship between *E. fulvus* and *S. spontaneum* and that the chromosome of *S. spontaneum* grew through rearrangement [52].

Previous studies demonstrated that *AP2/ERF* family genes play a vital role in abiotic stress responses and hormonal signaling regulation [36]. For example, in *Arabidopsis*, the three tandemly arranged *CBF/DREB* genes, *CBF1/DREB1B*, *CBF2/DREB1C*, and *CBF3/DREB1A*, are required for cold acclimation and freezing tolerance [53,54]. The ICE1/2 TFs positively regulate the expression of *CBF1/2/3* genes by binding to the MYC cis-acting element (CANNTG) of the *CBF1/2/3* promoter under cold stress, thus promoting *COR* gene expression and enhancing freezing tolerance in *Arabidopsis* [55]. In this study, we found that many *EfAP2/ERF* genes have MYC-binding site elements in their promoters (Figure 8), suggesting that the ICE-DREB/CBF-COR signaling cascade may play an important role in the cold stress response of *E. fulvus*. In addition, the DRE cis-acting element, Myb-binding site, and WRKY-binding site elements were also common among these genes, indicating that these TFs may participate in the regulation of *EfAP2/ERF* gene expression [56]. As signaling molecules, ABA and MeJA are involved in the regulation of *AP2/ERF* gene expression and adaptation to abiotic stress in plants (such as cold and drought tolerance) [18,57,58,59]. We found that almost all of the *EfAP2/ERF* genes have ABA and MeJA response elements in their promoters, suggesting that ABA and MeJA signaling may be involved in the adaptive response to stress in *E. fulvus*.

In recent years, some studies have also revealed that the cold tolerance function of *DREB/CBF* genes depends on interactions with other protein factors [60,61]. For example, overexpression of the *PbeNAC1* gene of *Pyrus betulifolia* in tobacco enhanced the cold tolerance and drought resistance of the transgenic plants, and further yeast two-hybrid (Y2H) and bimolecular fluorescence complementation (BiFC) analyses showed that the PbeNAC1 protein could physically interact with PbeDREB1 and PbeDREB2A [60]. In addition, in *Arabidopsis*, thioredoxin h2 (Trx-h2), a cytosolic redox protein, interacts with CBFs, changing the structure of CBF proteins to form CBF monomers, thus activating the expression of its target gene *COR* and increasing the frost resistance of *Arabidopsis*. Trx-h2 mutation can lead to increased sensitivity to cold stress. This suggests that the cold tolerance function of *CBFs* requires interactions with Trx-h2 proteins [61]. In this study, we found that 28 EfAP2/ERF proteins and 5 other proteins were mapped to the interaction network, suggesting their potential interactions (Figure 9). The results also suggest that the functions of some EfAP2/ERF proteins may depend on interactions with other proteins. However, future studies are required to reveal the function and interaction of these proteins.

It has been revealed that the AP2/ERF family TFs DREB/CBF play crucial roles in the response to cold and drought stress in *Arabidopsis* and rice [7]. In *Arabidopsis*, the expression of three *CBF/DREB* genes, *DREB1A/CBF3*, *AtDREB1B/CBF1*, and *AtDREB1C/CBF2,* was rapidly induced by cold stress, whereas the expression of *DREB2A* and *DREB2B* was induced by drought stress [62]. Similarly, in rice, the expression levels of *OsDREB1A/1B/1C* peaked at 6 h after cold treatment, but *OsDREB1G* expression started to increase at 6 h and peaked at 24 h [38]. *OsDREB2A* expression was markedly induced by drought [63], but *OsDREB2B* expression was induced by cold and drought stresses [64]. In previous studies, we identified the *EfDREB1A* gene induced by cold stress in *E. fulvus*. The *EfDREB1A* gene is homologous to the *DREB1A* gene of sorghum and rice [27] and is classified as the *EfDREB11* gene in this study. This study further confirmed that this gene was a cold stress-inducible gene, suggesting that it may have a cold response function. In addition, some studies have shown that *AP2/ERF* family genes have spatiotemporal expression specificity in different growth and development stages and tissues in plants [8,10]. However, no studies have investigated the cold stress response characteristics of *AP2/ERF* family genes in different tissues. In the present study, RNA-seq analysis showed that the expression levels of 16 *EfDREB* genes, 28 *EfERF* genes, and 6 *EfAP2* genes were upregulated by cold stress induction in leaves, roots, and stems, respectively (Figure 10), indicating that these genes are widely involved in the cold stress response and exhibit tissue-specific expression.

In *Arabidopsis* and rice, *DREB1s* and *DREB2s* are considered to be mainly responsive to cold stress and drought stress, respectively [38,62]. However, in *S. spontaneum*, some *DREB1s* (*SsDREB1F* and *SsDREB1L*) and *DREB2s* (*SsDREB2D* and *SsDREB2F*) respond to both cold and drought stresses [65]. In this study, further RT-qPCR verification revealed that the expression of seven *EfAP2/ERF* genes (*EfDREB10*, *EfDREB11*, *EfDREB39*, *EfDREB42*, *EfDREB44*, *EfERF43*, and *EfAP2-13*) was significantly upregulated under 72 h of cold stress (Figure 11), suggesting that these genes may have critical roles in the cold stress adaptation of *E. fulvus*. However, except for *DREB42*, the expressions of the other six genes were not induced by drought stress (Figure 12). In contrast, *EfDREB6* expression was upregulated under 9 d of drought stress but not induced by cold stress. These results indicated that two separate signal transduction pathways regulated the *EfAP2/ERF* genes under low-temperature and dehydration conditions. In addition, as in *Arabidopsis* and rice, two A1 group genes (*EfDREB10* and *EfDREB11*) in *E. fulvus* were induced by cold stress [38,62]. However, unlike *Arabidopsis*, two A2 group genes (*EfDREB16* and *EfDREB39*) in *E. fulvus* were not induced by drought stress [62]. In general, cold and drought stress often lead to the accumulation of ABA in plants [66]. Many studies have indicated that *DREB/CBFs* participate in the response to abiotic stress via both ABA-dependent and ABA-independent pathways [33,34]. Furthermore, we found that *EfDREB5* was upregulated, four *EfAP2/ERF* genes (*EfDREB11*, *EfDREB39*, *EfERF43*, and *EfAP2-13*) were downregulated, and four *EfDREB* genes (*EfDREB10*, *EfDREB16*, *EfDREB42*, and *EfDREB44*) showed no significant up- or downregulation after ABA treatment (Figure 13). These results suggest that *EfAP2/ERF* genes might participate in the response to abiotic stress via ABA-dependent and ABA-independent pathways.

A previous study showed that *E. fulvus* had strong cold resistance. Most notably, this study found that the expression of most *EfAP2/ERF* genes was induced by cold stress. Whether some *EfAP2/ERF* genes contribute to cold tolerance in *E. fulvus* still needs to be confirmed by further studies. Overall, this study not only lays the foundation for further revealing the function and expression regulation mechanism of *EfAP2/ERF* family genes but also provides valuable reference genes for the genetic improvement of sugarcane.

## 4. Materials and Methods

### 4.1. Identification and Classification of AP2/ERF Family Genes in E. fulvus

The genome sequences and sequence information of *E. fulvus* were downloaded from the *Erianthus fulvus* Genome Database (EfGD) (http://efgenome.ynau.edu.cn/ (accessed on 5 October 2021)) [41]. Gene family identification and analysis were performed with the Docker image tool (OmicsClass/Gene-family V1.0.1, builed by OmicsClass (Beijing, China)). The HMM profile of the AP2 domain (PF00847.21) was downloaded from the Pfam database (http://pfam.xfam.org/ (accessed on 15 October 2021)) and used for identification of AP2/ERF genes in HMMER with a cut-off E-value ≤ 0.01 for domain screening. The AP2/ERF superfamily proteins of *S. spontaneum* [8] were used as query sequences in the local BLAST program to find members of AP2/ERF superfamily genes of the *E. fulvus* genome with the following parameter: expected values ≤ 0.01. The two-part genes were combined as candidate AP2/ERF genes. In addition, Pfam (http://pfam.xfam.org/ (accessed on 28 October 2021)), NCBI-CDD (https://www.ncbi.nlm.nih.gov/cdd (accessed on 28 October 2021)), and SMART (http://smart.embl-heidelberg.de/ (accessed on 28 October 2021)) searches were performed to identify AP2/ERF members. ExPASy (https://web.expasy.org/protparam/ (accessed on 10 January 2022)) was used to predict AP2/ERF transcription factors based on MW and pI.

### 4.2. Phylogenetic, Conserved Motif, and Gene Structure Analyses of E. fulvus AP2/ERF Genes

Multiple sequence alignment of AP2/ERF family proteins was performed using Clustal X v2.1 with the default parameters. An unrooted neighbor joining (NJ) tree with 1000 bootstrap replications was constructed using MEGA 7.0 based on full-length protein alignment. Conserved motifs were predicted using the MEME Suite web server (http://meme-suite.org/ (accessed on 24 February 2022)) with the following parameters: maximum number of motifs set at 25 and expected values ≤ 0.001. The structural information of *AP2/ERF* family genes was obtained based on the annotation file (gff3) of the *E. fulvus* genome. Finally, we used TBtools software to integrate phylogenetic trees, conserved motifs, and gene structure results.

### 4.3. Chromosomal Distribution, Gene Duplication, and Synteny Analysis of EfAP2/ERF Family Genes

The chromosomal distribution information of the *AP2/ERF* genes was extracted from the *E. fulvus* genome annotation GFF3 file, and the results obtained were visualized using MG2C_v2.1 (http://mg2c.iask.in/mg2c_v2.1/ (accessed on 6 March 2022)). Analysis of gene duplication events was performed using the Multiple Collinearity Scan toolkit (MCScanX) [29]. Dual Synteny Plotter software (https://github.com/CJ-Chen/TBtools/ (accessed on 26 March 2022)) was used to determine the syntenic relationship of the *AP2/ERF* genes from *E. fulvus* and other selected plants. Tandem duplications and segmental duplications were identified using the method described by Li et al. [8]. Nonsynonymous (ka) and synonymous (ks) substitutions of each duplicated AP2/ERF gene were calculated using KaKs_Calculator 2.0 [67]. The divergence time (T) was calculated by T = Ks/(2×6.1×10 − 9)×10 − 6 Mya [8].

### 4.4. Identification of Cis-Acting Regulatory Elements in Promoters of EfAP2/ERF Family Genes

According to the chromosomal location information of *AP2/ERF* family genes from the GFF3 file, 2 kb genomic sequences upstream of the transcriptional start site of each *AR2/ERF* gene were extracted as the promoter and then submitted to the plantCARE website (http://bioinformatics.psb.ugent.be/webtools/plantcare/html/ (accessed on 3 April 2022)) to identify possible cis-acting regulatory elements [68]. Moreover, we used TBtools software to visualize the results identified.

### 4.5. Protein Network Interaction and Gene Expression Analysis

According to the method described by Lei et al. [69]. OrthoVenn2 (https://orthovenn2.bioinfotoolkits.net/home (accessed on 4 May 2022)) software was used to find orthologous pairs between EfAP2/ERF and AtAP2/ERF proteins. Then, the interaction networks of EfAP2/ERF proteins were identified based on the orthologous genes between *E. fulvus* and *Arabidopsis* using STRING (https://cn.string-db.org/ (accessed on 4 May 2022)) software and visualized using Cytoscape v3.8.2 (https://cytoscape.org/ (accessed on 7 May 2022)) software. To identify the expression pattern of *EfAP2/ERF* genes, the RNA-seq expression data (FPKM value) of *E. fulvus* ‘99-1’, including three tissues (root, stem, and leaf) under low temperature (4 °C) stress (0, 24, and 72 h), were obtained from the EfGD [41]. The heatmap of *EfAP2/ERF* genes was constructed using TBtools software.

### 4.6. Plant Material, Stress Treatment, and RT-qPCR Analysis

*E. fulvus* ‘99-1’ clones from the Sugarcane Research Institute of Yunnan Agricultural University were planted in pots and cultured in a greenhouse until the seedling stage with 4–5 leaves. For cold treatment, seedlings were moved to a low-temperature light incubator with a light intensity of 300 μmol/(m^2^·s), light cycle of 12/12 h (light/dark), 65% humidity, and low temperature (4 °C). Then, the leaves (the first fully expanded leaf from top to bottom) were harvested at 0 h (CK), 24 h, and 72 h. For drought treatment, seedlings were not watered, and then leaves (the first fully expanded leaf from top to bottom) were collected at 0 d (control, water content: 70 ± 5%), 3 d (water content: 50 ± 5%), 6 d (water content: 30 ± 5%), and 9 d (water content: 15 ± 5%) [70]. For ABA treatment, tissue culture seedlings of *E. fulvus* ‘99-1’ were sprayed with abscisic acid (ABA, 100 μM), and then leaves were collected at 0 h (CK), 6 h, and 12 h [8]. The total RNA of leaves was extracted using the TRIzol kit (Tiangen, Beijing, China), following the instructions of the manufacturer. Then, cDNA was synthesized by reverse transcription of 1 μg of RNA using the Fast Quant RT Super Mix kit (Tiangen, Beijing, China). Subsequently, real-time quantitative PCR (RT-qPCR) was carried out using the SuperReal PreMix Plus (SYBR Green) kit (Tiangen, Beijing, China) on an ABI 7500 fluorescence quantitative PCR instrument. The gene-specific RT-qPCR primers (Appendix A) were designed using Primer5 software, and the *25S-rRNA* gene was selected as an internal reference gene to normalize the gene expression levels. The reaction system and RT-qPCR procedure are shown in Appendix A. Each treatment was replicated 3 times. The expression levels of *EfDREB* genes were calculated by the 2^−ΔΔCT^ method.

### 4.7. Statistical Analysis

We performed three independent biological duplicates in each experiment. GraphPad Prism 8 software (version v8.0.1.244) was used to perform statistical analyses. The data are presented as the means, with error bars representing standard deviations (mean ± SD). The significant differences relative to controls (one-way ANOVA with Tukey’s multiple range test) are indicated by * *p* < 0.05, ** *p* < 0.01, *** *p* < 0.001, and **** *p* < 0.0001.

## 5. Conclusions

In this study, we identified 145 *EfAP2/ERF* family genes in the whole genome of *E. fulvus*, carried out sequence analysis, phylogenetic analysis, gene structure analysis, evolutionary analysis, and protein interaction analysis, and elaborated the structural characteristics, potential functional characteristics, and possible evolutionary mechanisms of the members of this gene family. Their expression patterns under low temperature, drought, and ABA treatment were studied. These results will be helpful in providing valuable resources to better understand the biological role of the *E. fulvus AP2/ERF* genes.

## Figures and Tables

**Figure 1 ijms-24-07102-f001:**
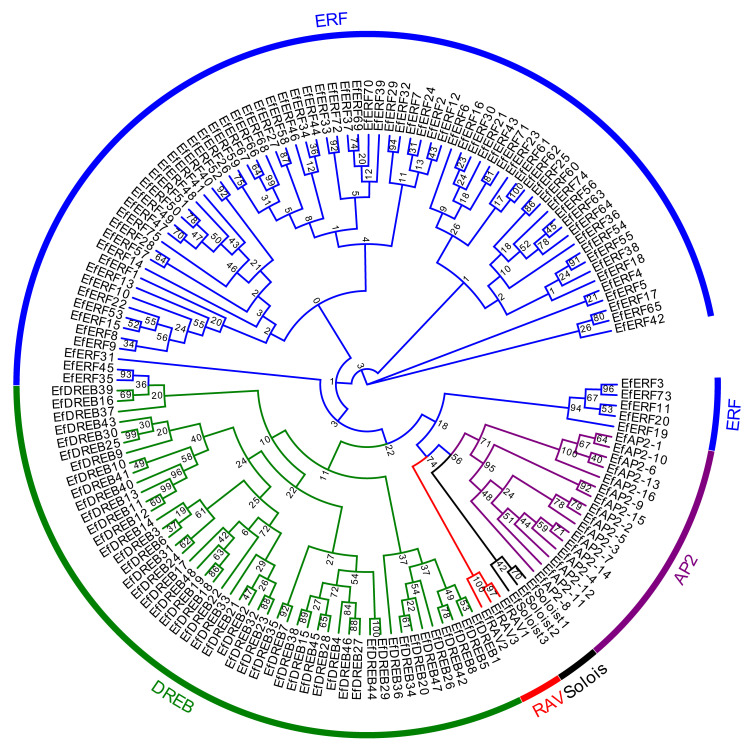
Phylogenetic tree of *AP2/ERF* family genes in *E. fulvus*. The phylogenetic trees clustered all of the EfAP2/ERF proteins into five subfamilies: the DREB, ERF, AP2, RAV, and Soloist subfamilies. Each subfamily is shown in a different color.

**Figure 2 ijms-24-07102-f002:**
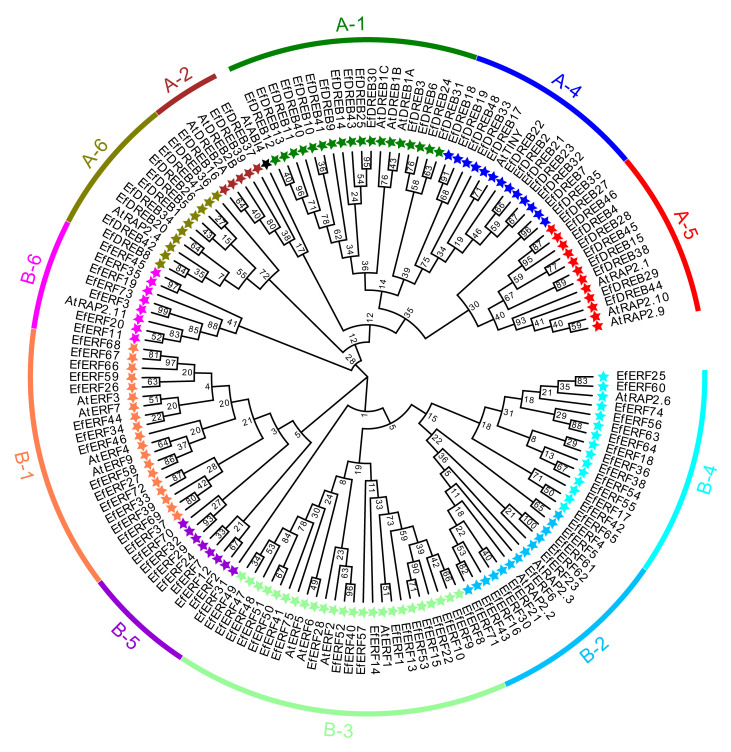
Phylogenetic tree of *DREB* and *ERF* subfamily genes in *E. fulvus* and *Arabidopsis*. The *EfDREB* and *EfERF* subfamily genes were divided into five groups (A1 to A2 and A4 to A6) and six groups (B1 to B6), respectively.

**Figure 3 ijms-24-07102-f003:**
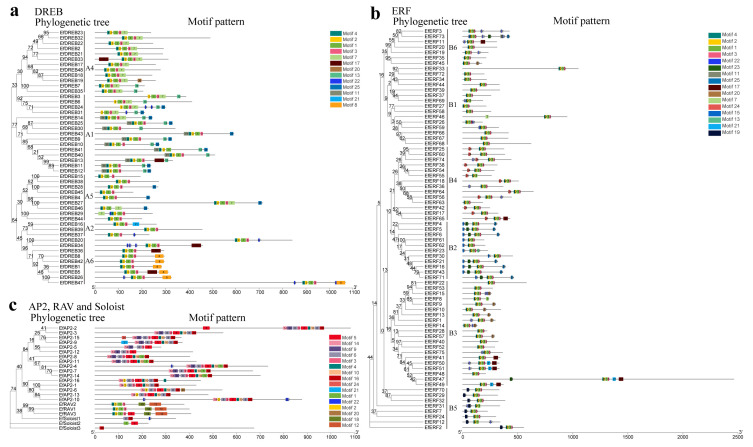
Conserved motif cluster analysis of *EfAP2/ERF* family genes according to phylogenetic relationships. (**a**) Conserved motif of *EfDREB* subfamily. (**b**) Conserved motif of *EfERF* subfamily. (**c**) Conserved motif of *EfAP2*, *EfRAV*, and *EfSoloist* subfamilies. A total of 25 motifs are displayed in different-colored boxes.

**Figure 4 ijms-24-07102-f004:**
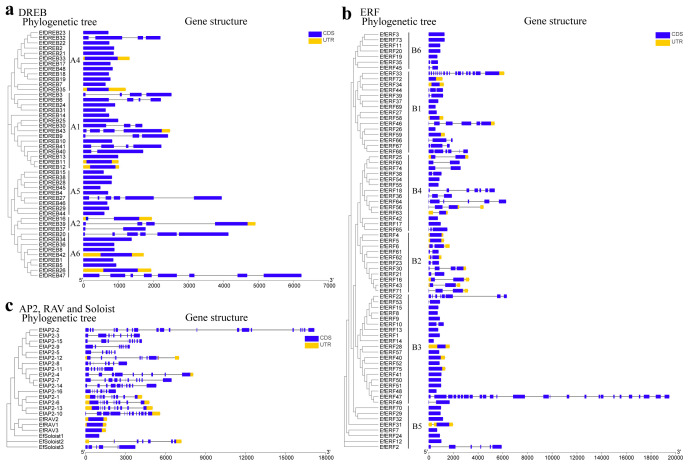
Gene structure cluster analysis of *EfAP2/ERF* family genes according to phylogenetic relationship. (**a**) Gene structure of *EfDREB* subfamily. (**b**) Gene structure of *EfERF* subfamily. (**c**) Gene structure of *EfAP2*, *EfRAV*, and *EfSoloist* subfamilies.

**Figure 5 ijms-24-07102-f005:**
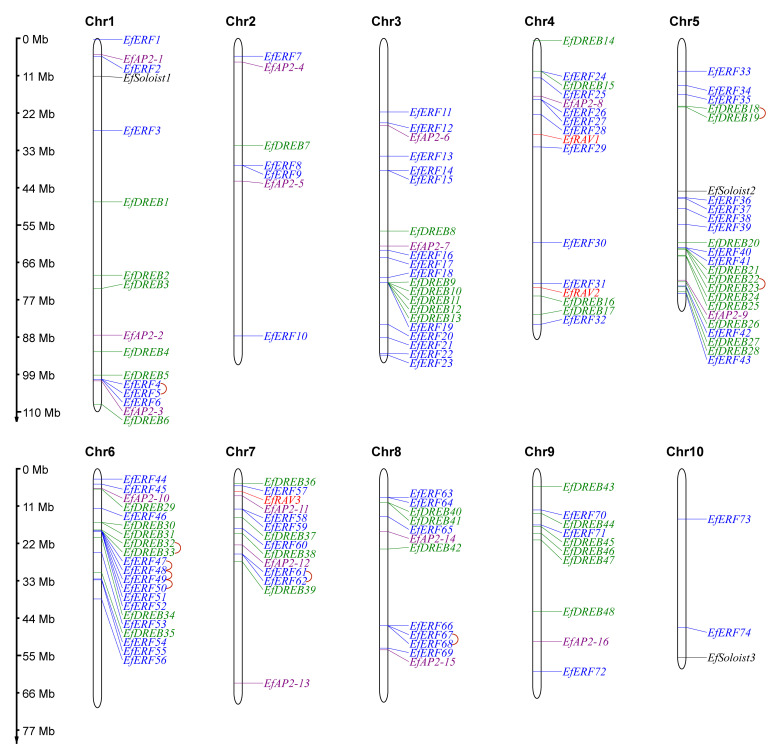
Chromosomal distribution and tandem duplication analysis of *EfAP2/ERF* family genes. A red line between two gene names indicates that they are tandem duplication gene pairs. Each subfamily gene name is shown in a different color.

**Figure 6 ijms-24-07102-f006:**
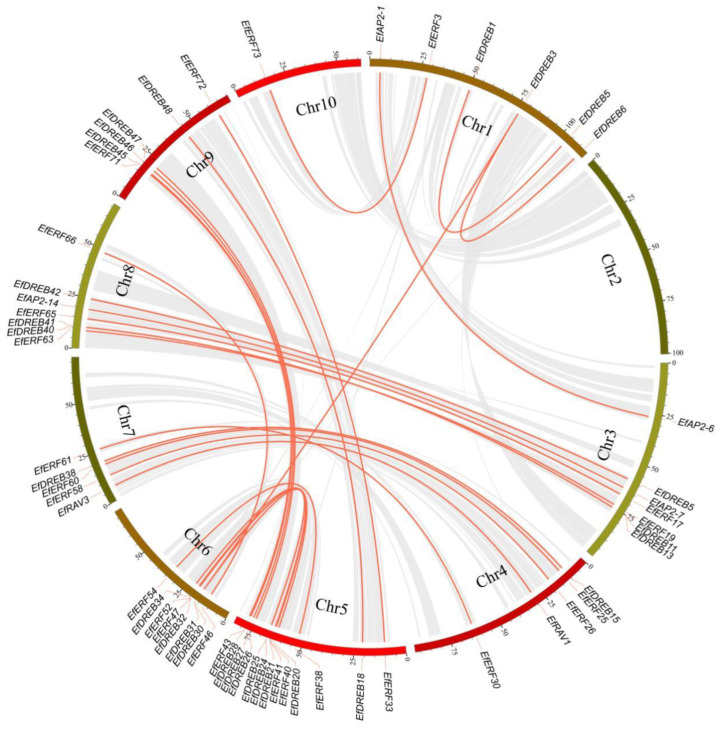
Interchromosomal segmental duplication gene pair analysis of the EfAP2/ERF family. A red line between two gene names indicates that they are segmental duplication gene pairs.

**Figure 7 ijms-24-07102-f007:**
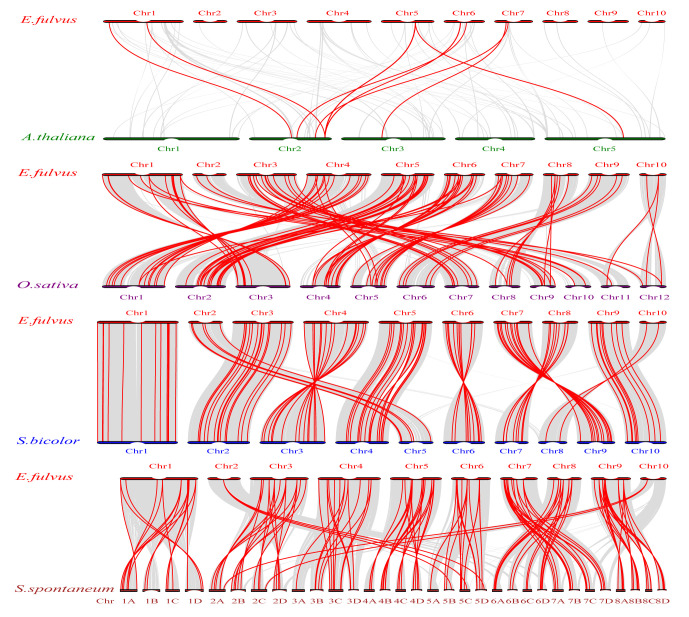
Synteny analysis of *AP2/ERF* genes among the representative plant species *E. fulvus*, *A. thaliana*, *O. sativa*, *S. bicolour*, and *S. spontaneum*. Grey lines in the background indicate the collinear regions within *Erianthus fulvus* and other plant genomes, while the red lines highlight the collinear *AP2/ERF* gene pairs.

**Figure 8 ijms-24-07102-f008:**
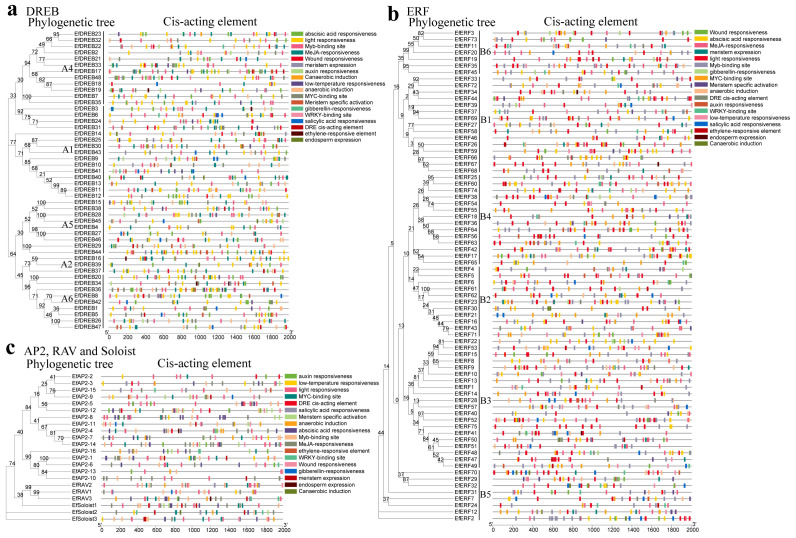
Promoter cis-acting element analysis of *EfAP2/ERF* family genes according to phylogenetic relationship. (**a**) Cis-acting element of *EfDREB* subfamily. (**b**) Cis-acting element of *EfERF* subfamily. (**c**) Cis-acting element of *EfAP2*, *EfRAV*, and *EfSoloist* subfamilies. Each cis-acting element is displayed in different-colored boxes.

**Figure 9 ijms-24-07102-f009:**
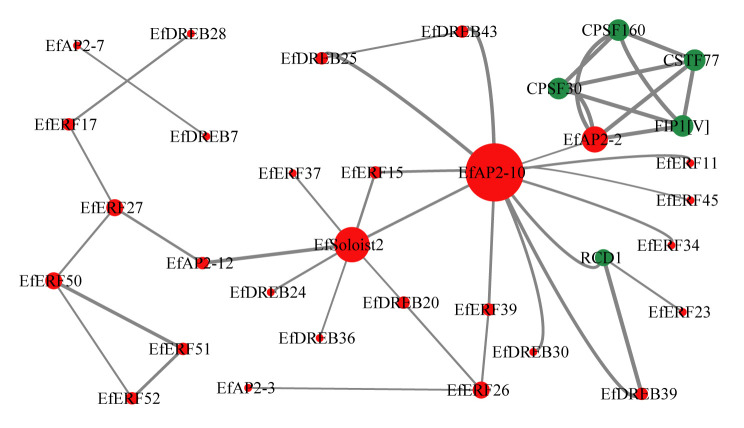
Interaction network of EfAP2/ERF proteins in *E. fulvus* according to orthologues in *Arabidopsis*. Each protein is displayed in different-colored circles, red indicating EfAP2/ERF family proteins, and green indicating other protein families.

**Figure 10 ijms-24-07102-f010:**
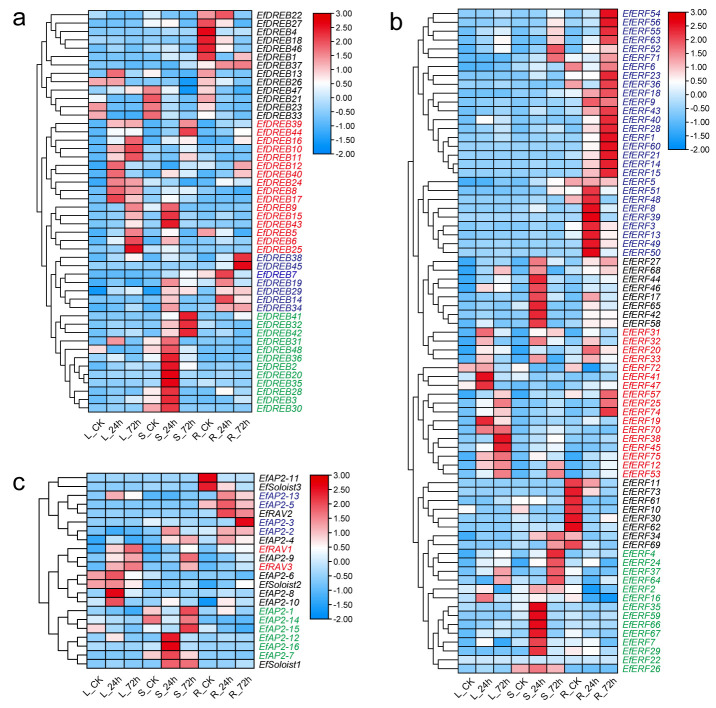
Expression profile of *EfAP2/ERF* genes in different tissues (leaves (L), roots (R), and stems (S)) and under low-temperature (4 °C) stress (0, 24, and 72 h). (**a**) Expression profile of *EfDREB* genes. (**b**) Expression profile of *EfERF* genes. (**c**) Expression profile of *EfAP2*, *EfRAV*, and *EfSoloist* genes. For each line, the expression patterns are presented as heatmaps in blue/white/red boxes, with red indicating high expression level, white indicating moderate expression level, and blue indicating low expression level. The red/green/blue gene names indicate that these genes were upregulated by cold stress induction in leaves, stems, and roots, respectively.

**Figure 11 ijms-24-07102-f011:**
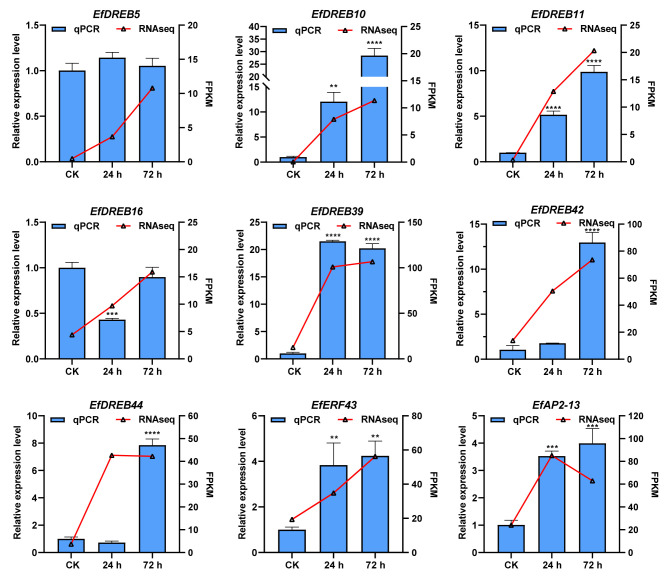
Expression profiles of 9 selected *EfAP2/ERF* genes in response to cold stress. The *25SRNA* gene was used as the internal control gene. Each column represents the mean of three independent replicates. The error bars represent the standard deviations of the mean (mean ± SD). The significant differences relative to controls (one-way ANOVA with Tukey’s multiple range test) are indicated by ** *p* < 0.01, *** *p* < 0.001, and **** *p* < 0.0001. Line charts represent FPKM values of the RNA-seq data.

**Figure 12 ijms-24-07102-f012:**
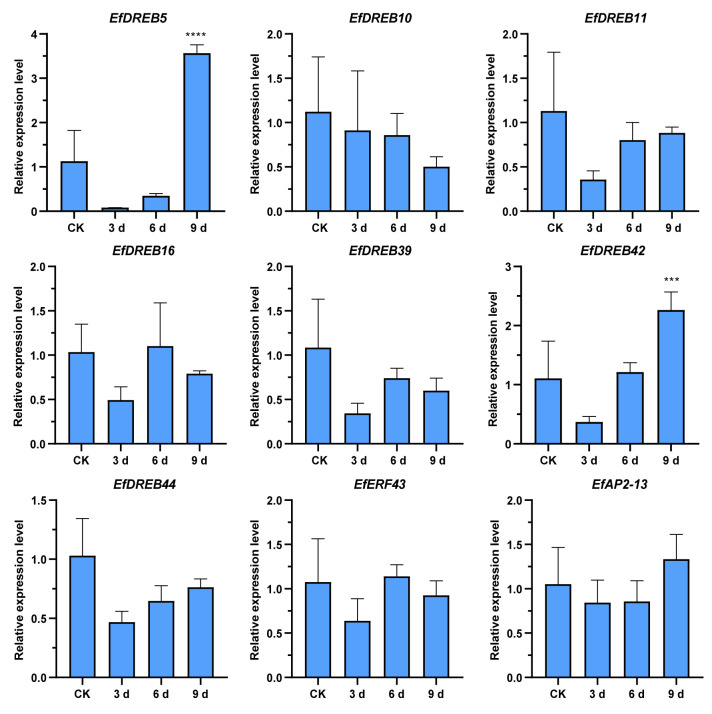
Expression profiles of 9 selected *EfAP2/ERF* genes in response to drought stress. Each column represents the mean of three independent experiments. The error bars represent the standard deviations of the mean. The significant differences relative to controls (one-way ANOVA with Tukey’s multiple range test) are indicated by *** *p* < 0.001, and **** *p* < 0.0001.

**Figure 13 ijms-24-07102-f013:**
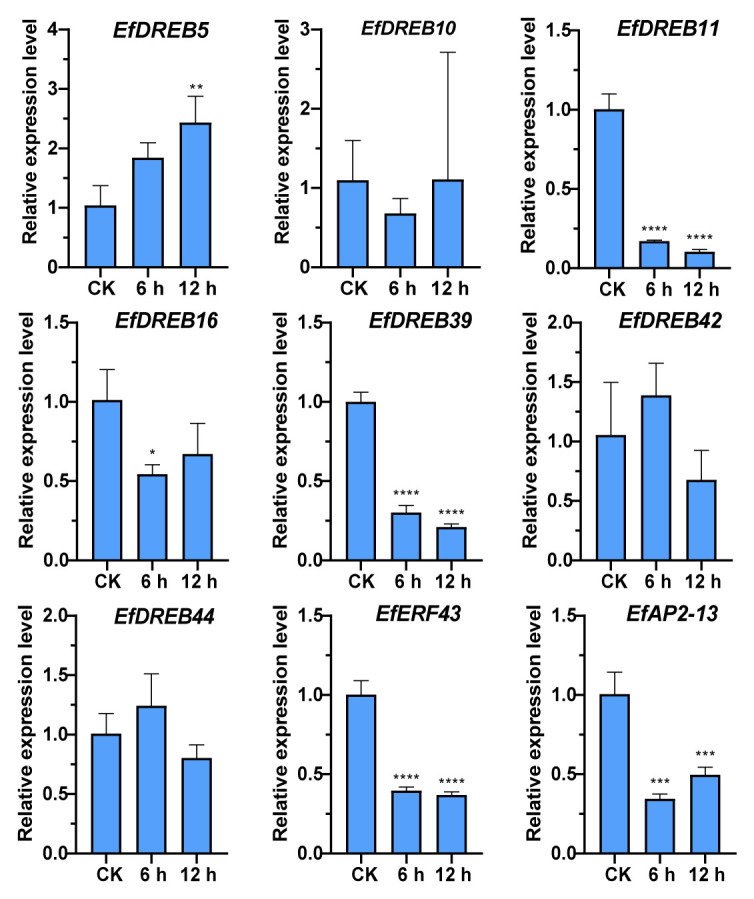
Expression profiles of 9 selected *EfAP2/ERF* genes in response to ABA treatment. Each column represents the mean of three independent experiments. The error bars represent the standard deviations of the mean. The significant differences relative to controls (one-way ANOVA with Tukey’s multiple range test) are indicated by * *p* < 0.05, ** *p* < 0.01, *** *p* < 0.001, and **** *p* < 0.0001.

## Data Availability

The data presented in this study are available in the article and Appendix A.

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
