# Peer review of "Genome-Wide Identification, Evolution, and Expression Analyses of AP2/ERF Family Transcription Factors in Erianthus fulvus"

_ijms, 2023, doi:10.3390/ijms24087102_

Round 1

Reviewer 1 Report

Line 203: Where did it show “A total of 11 EfAP2/ERF genes showed synte11 EfAP2/ERF genes with……” In the figure, is the number 8?

Sections 2.9 and 2.10: Move the results of cold stress in section 2.10 to section 2.9. Revise the corresponding titles of these sections.

Reviewer 2 Report

CONGRATS VERY GOOD PAPER

Author Response

Response to Reviewer 2 Comments

comment: CONGRATS VERY GOOD PAPER

Response: We are very grateful to the reviewer. Your high comments on the manuscript deeply inspired our plan for further work. This manuscript is an important stepping stone. Based on this manuscript, we will further study the overexpression, gene editing and expression regulation mechanisms of related genes. We thank you again.

Reviewer 3 Report

Low temperature and drought are the main environmental factors affecting sugarcane growth, yield and quality. Therefore, mining sugarcane germplasm resources of cold-tolerant and drought-resistant to provide parents or donor genes for sugarcane breeding has become the main focus of stress-resistant sugarcane breeding. However, the mechanisms of cold tolerance and drought re-sistance in E. fulvus are still unclear. The EfDREB10, EfDREB11, EfDREB39, EfDREB42, EfDREB44, EfERF43 and EfAP2-13 responded to cold stress, EfDREB5 and EfDREB42 responded to drought stress, and EfDREB5, EfDREB11, EfDREB39, EfERF43 and EfAP2-13 responded to ABA treatment. These results will be helpful for better understanding the molecular feature and biological role of the E. fulvus AP2/ERF genes and lay a foundation for further research the function of EfAP2/ERF genes and the regulatory mechanism of the abiotic stress response.

However, some minor comments to this paper.

1.     There are some minor mistakes in the references

2.     The part of discussion is not well discussed combined with results in this study and previous studies of EfAP2/ERF genes with cold and drought response.

3.     The interaction network of EfAP2/ERF proteins need to be verified by the bench work.

Round 2

Reviewer 3 Report

The revised version has addressed all the comments. The papper can be acceptted as it is.